# Perceptual Biases as the Side Effect of a Multisensory Adaptive System: Insights from Verticality and Self-Motion Perception

**Luigi F. Cuturi** [1,2]

1    Department of Psychology, Università di Torino, 10124 Turin, Italy; luigifelice.cuturi@unito.it
2    Unit for Visually Impaired People, Istituto Italiano di Tecnologia, 16163 Genoa, Italy

**Abstract:** Perceptual biases can be interpreted as adverse consequences of optimal processes which otherwise improve system performance. The review presented here focuses on the investigation of inaccuracies in multisensory perception by focusing on the perception of verticality and self-motion, where the vestibular sensory modality has a prominent role. Perception of verticality indicates how the system processes gravity. Thus, it represents an indirect measurement of vestibular perception. Head tilts can lead to biases in perceived verticality, interpreted as the influence of a vestibular prior set at the most common orientation relative to gravity (i.e., upright), useful for improving precision when upright (e.g., fall avoidance). Studies on the perception of verticality across development and in the presence of blindness show that prior acquisition is mediated by visual experience, thus unveiling the fundamental role of visuo-vestibular interconnections across development. Such multisensory interactions can be behaviorally tested with cross-modal aftereffect paradigms which test whether adaptation in one sensory modality induces biases in another, eventually revealing an interconnection between the tested sensory modalities. Such phenomena indicate the presence of multisensory neural mechanisms that constantly function to calibrate self-motion dedicated sensory modalities with each other as well as with the environment. Thus, biases in vestibular perception reveal how the brain optimally adapts to environmental requests, such as spatial navigation and steady changes in the surroundings.

**Keywords:** bias; vestibular; graviception; verticality; self-motion; aftereffect

## 1. Introduction

When moving through space, our aim is to accomplish a specific goal-directed action, such as reaching the door entrance of a building. To finalize this action, our brain processes all relevant sensory information to build a reliable perceptual representation of physical properties of interest. For instance, the visual system will identify environmental properties, such as the location of the entrance door to the building. This process provides the basis for action to finalize the goal, that is to spatially navigate until we reach the door. While moving, our actions need to be calibrated relative to our perception of the environment. Specifically, in the case of spatial navigation, perception changes as we move through space. We experience a compound of multisensory information that provides our brain with the cues necessary to accomplish our action, e.g., visual (i.e., optic flow), vestibular (i.e., self-motion), haptic/tactile (i.e., podotactile readout of the floor texture) and auditory (i.e., acoustic flow and landmarks). This combination of cues builds a dynamic perceptual representation of the environment as we move through. Although some information has been predefined before performing the action (e.g., the door's location and the path to be performed), the environment is subject to changes (e.g., the sudden arrival of another agent who crosses our path) and we adapt our action to such changes. Thus, our action needs to be calibrated relative to the perceptual properties of stimulus readout, whereas perception changes as we act in the environment. It follows that such operations mutually influence each other leading to perceptual and motor calibration.

From the above presented spatial navigation example, it emerges that perceptual accuracy is fundamental to accomplish actions in the environment and that it is more than the simple product of sensory readout. In order to study this rather complex phenomenon, psychophysical studies take advantage of biases, that is inaccuracies in perception. Such perceptual errors reveal how the brain processes sensory information and permit to infer properties of the brain leading to such inaccuracies. A clear representation of the importance of biases as a side effect of an optimal perceptual system can be observed in Figure 1 where an overestimation symmetric around a category boundary is represented.

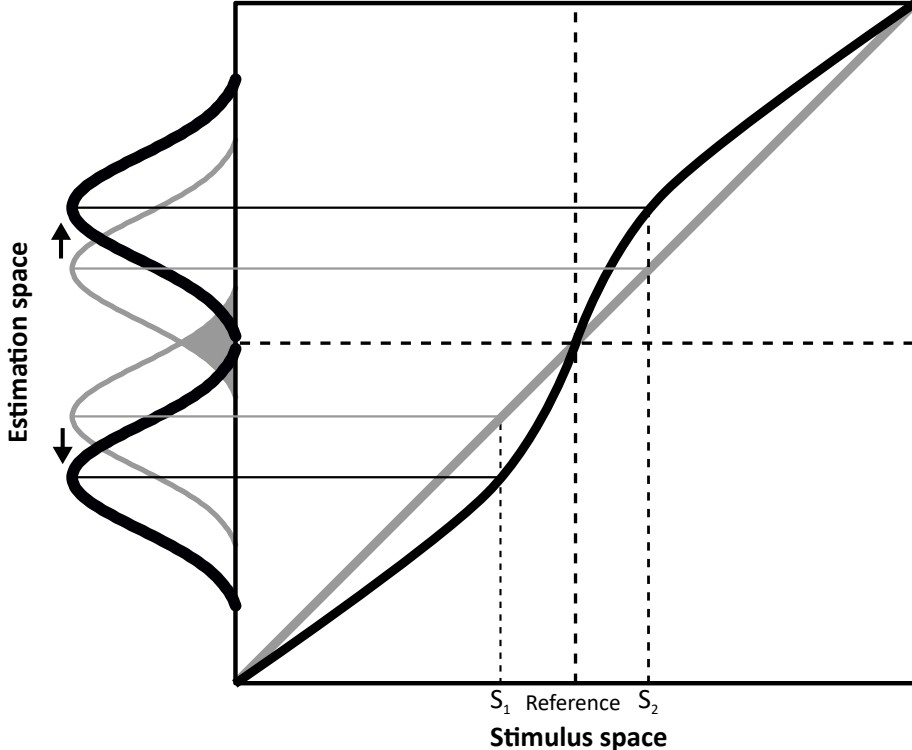

**Figure 1.** Representation of unbiased (in gray) and biased (in black) perceptual systems. The diagonal unbiased gray line shows a system where estimates are identical to the stimulus value. Biases in the thick black line are shown by the deviation of the estimates from the diagonal which is reflected in a repulsion from the reference. Figure is inspired by [1].

Given this scenario, it can be postulated that discriminability relative to the reference defined by the category boundary would be higher with respect to a system where perceptual space is uniformly distributed relative to the stimulus space. In detail, Figure 1 shows two hypothetical perceptual systems, one that is unbiased (in gray) and another biased system (in black). In both, estimates are represented as a probability distribution (i.e., Gaussian). Peaks indicate the system's estimates of the stimuli (e.g., $S_1$ and $S_2$) and width (in terms of standard deviation) represents variability. This representation assumes that physical stimuli are grouped in two opponent categories (e.g., left and right) separated by a reference located at the center of the stimulus space (i.e., the category boundary). If the perceptual system is unbiased (in gray), estimates will be equal to the stimulus values, i.e., all data points fall on the diagonal. However, because of perceptual variability, overlaps around the category boundary could lead to category misclassification (e.g., misinterpretation of a rightward tilted line as leftward oriented, and thus belonging to the wrong category) as indicated by the gray area between the two Gaussian describing unbiased estimates for $S_1$ and $S_2$. On the contrary, in the biased system (in black), estimates are repelled from the reference thus showing overestimation. Assuming the same perceptual variability as for the unbiased system, overlaps across the category boundary are reduced in the biased system because of the estimates' shift ($S_1$ and $S_2$ are overestimated). Therefore,

discriminability in terms of category classification is improved as the system benefits from the increased separation between the estimates to tell their relevant category. In other words, the ability to discriminate between two different stimuli (e.g., $S_1$ and $S_2$) as belonging to two different categories (e.g., left/right) defined by the category boundary would be aided by the overestimation represented in Figure 1.

From the previously mentioned example of spatial navigation, it is clear that not only does our representation of the physical world influence perception, including idiosyncratic differences across individuals [2], but contextual information (either spatially or temporally defined) also affects perceptual reading of the environment where our action takes place. For instance, in the case of constant novel stimulation, the brain reduces its response to continuous stimulation thus enhancing detection of changes in the new state-of-the-world. This is clearly experienced with the light/dark adaptation, where after long exposure to a darkened environment, visual sensitivity to light is increased to maintain optimal visual discrimination. In this case, the nervous system adapts the perceptual baseline to a novel steady stimulation (i.e., reduced light) to recalibrate visual processing in response to the novel environmental properties. Similarly, prolonged roll-tilt of the head on a side induces shifts in the perception of verticality even after returning to the upright position [3–5]. In laboratory settings, byproducts of brain's adaptability are revealed by biases induced through sustained exposure to steady stimulation. Recalibration of the perceptual baseline leads the brain to adapt to the novel context thus improving discriminability [1,6–9]. However, as a consequence, neutral or ambiguous stimulation presented after adaptation can be misinterpreted by the brain.

Therefore, biases can be considered as the price of optimality for improving other aspects of system performance (e.g., detection of changes in the environment). Such discrepancies between physical properties of the stimulus and the estimates computed by our brain can be interpreted here as the side effect of a system that operates optimally in two main ways: (1) it avoids misclassification errors by reducing the effect of sensory noise on the readout of stimuli close to the category boundary; (2) it adapts to the changes in the environment by reducing coding redundancy.

Often considered as the sixth sense, the vestibular system allows the brain to compute information, such as the direction of gravity and self-motion properties. This system is composed by biomechanical sensors, that can be considered as linear and angular accelerometers that provide information about head movement. The peripheral vestibular organs comprise two main types located in the inner ear in the temporal bones. These are the semicircular canals and the otolith organs (see Figure 2). The three canals respond to angular head movements while the otoliths signal linear head movements due to translation and gravity. In both organs, the transduction of movement into electrochemical signals is accomplished by hair cells whose polarization depends on deflection of their cilia. Inside the canals, such cells are grouped in a structure called the ampulla. The canal is filled with a viscous liquid called endolymph. As the head rotates, the ampulla is moved because of the endolymph lagging behind the duct walls and pressing on the cupula, which is the gelatinous structure that covers the ampulla. The hair cells then change their firing rate accordingly and the signal is sent to the vestibular nuclei in the brainstem. In each ear there are three semicircular canals approximately orthogonal to each other. The combination of signals sent by the semicircular canals provides the information necessary to sense direction and speed of angular movement. In particular, when the head is upright in the earth-horizontal plane, the horizontal canal mostly signals yaw rotations, whereas anterior and posterior canals respond to combinations of pitch and roll rotations (see Figure 2A). Translational movements are processed via the otolith organs whose functioning is similar to the semicircular canals. The utricle is sensitive to the upright head's movements on the horizontal plane, while the sacculus responds to vertical movements on the sagittal plane. Inside the otolith organs, the cilia of the hair cells are embedded in a gelatinous layer covered with calcium carbonate crystals, namely otoconia, providing the inertial force that reflects head movement. As the head moves linearly, the weight of the otoconia produces a

shearing force on the underlying hair cells causing them to bend (see Figure 2B). In both otolith organs, the hair cells are embedded in a sensory epithelium called the macula. Each hair cell has one kinocilium and 20 to 100 stereocilia. As the cilia bend the cell's firing rate changes depending on its polarization (defined by the position of the kinocilium, see Figure 2B). Each individual hair cell can therefore indicate acceleration or deceleration in only one direction, but taking all hair cells together, all movement directions are represented in the plane of each macula. In this way, the combination of semicircular canal and otolith activation allows transduction of natural movements of the head including both angular and linear components. The vestibular system thus provides an independent cue for self-motion perception and graviception that aids perception when other sensory modalities informing about the environment (e.g., visual or auditory) are corrupted or unavailable.

From the vestibular peripheral organs, the signal is directed to the vestibular nuclei in the brainstem. From this site, inertial information is relayed to and processed by several brain structures to maintain posture (projections from the vestibular nuclei to the spinal cord through the vestibulo-spinal reflex) and balance, gaze fixation by compensating head movements through the vestibular-ocular reflex (from the vestibular nuclei to motor neurons in the brainstem) and finally body position (connections from the vestibular nuclei to cerebellar regions and thalamocortical pathways) and self-motion perception. In the latter case, second order neurons in the brainstem project to thalamic nuclei which project to cortical regions, such as the posterior insula, superior temporal gyrus, inferior parietal lobule (angular and supramarginal gyrus), somatosensory cortex, precuneus, cingulate gyrus, frontal cortex (motor cortex and frontal eye fields), and hippocampus (for a review, see [10]).

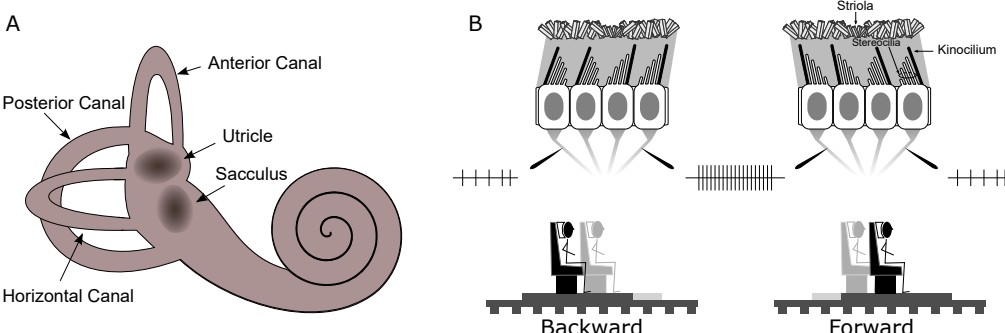

**Figure 2.** (**A**) The peripheral vestibular system. Pictorial representation of the semicircular canals and the otolith organs (utricle and sacculus). (**B**) Activation of the neural afferents in response to a backward (left column) and forward (right column) movement. In the top row, the physiological response is represented: the afferents increase their firing rate only when the stereocilia bend toward the Kinocilium. Oppositely tuned cells are separated by a section of the macula with smaller otoconia called the striola. In the bottom row, movements are represented as they can be elicited in the laboratory with a motion platform as in [11,12].

In the investigation of the vestibular system and perception based on vestibular signals in particular, much research has employed psychophysical methods to investigate the interaction of sensory modalities such as vision, haptic and audition with vestibular. In the context of this review, studies on vestibular perception and its interaction with the other sensory modalities will be presented to elucidate the role of biases in perceptual processing.

## 2. Inaccuracies in the Perception of Verticality

In situations in which not all sensory modalities are available, the vestibular system may be vulnerable to biases and limitations that reduce its reliability in perceptual judgement. When moving linearly on the earth-horizontal plane while upright, e.g., while driving a car or flying an airplane, the vestibular system codes self-motion as the combination of gravity acting vertically and inertial acceleration acting horizontally on the sensors. Due to Einstein's principle of equivalence, the otolith organs within the vestibular system

are unable to distinguish these two forces. Instead, the combined gravitoinertial vector is sensed (see Figure 3).

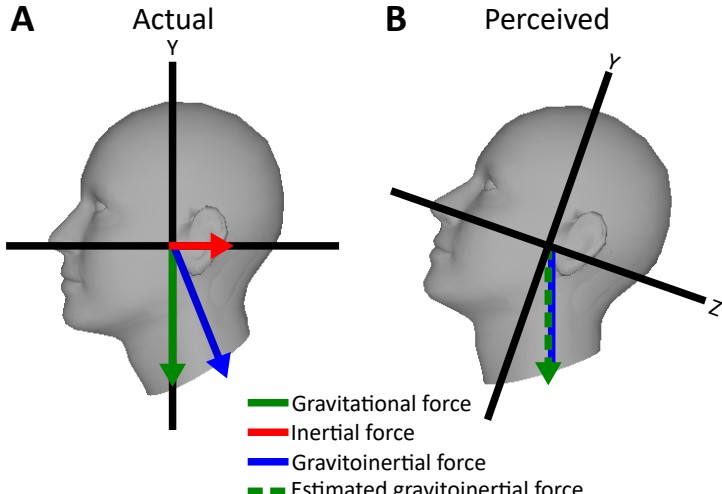

**Figure 3.** The somatogravic illusion. (**A**): Inertial (red arrow) and gravitational (green arrow) forces are combined in vector sum indicating gravitoinertial force (blue arrow). (**B**): When cues that help decomposing the vector sum are absent, gravitoinertial force is interpreted to be due to the gravitational force (superimposed green and blue arrows) leading to false head pitch perception. Figure adapted from [13].

In order to distinguish the two forces, the brain takes advantage of other sensory information such as the angular velocity signals from the semicircular canals, and environmental information coded via the other senses (e.g., the optic flow). In situations where most informative sensory modalities such as vision are unreliable, and rotational components of motion are weak, illusions might occur. For instance, a well-known issue in aviation is the so-called somatogravic illusion in which strong sustained linear acceleration can be misperceived leading to illusory head pitch [14,15] (see Figure 3). While flying through a cloud bank or a storm, visual information might be highly corrupted and unable to provide additional sensory information to the ones sensed by the vestibular system. Therefore, the brain might be unable to distinguish between gravitational and inertial force. Thus, combined gravitoinertial force acting on the vestibular system might be erroneously interpreted as to be the gravitational force perpendicular to the earth horizontal plane, inducing a biased perception of tilt (see Figure 3). This phenomenon can cause the erroneous correction of flight trajectory that leads to lowering the airplane's pitch, leading to increased acceleration with a high risk of plane crash [14].

Perception of verticality indicates the ability to correctly process the direction of gravity, in other words graviception. In the context of vestibular perception and the assessment of balancing abilities, the ability of perceiving the direction of gravity is fundamental to maintain balance. Indeed, such an ability is fostered by the ability to detect any deviation from the gravitational axis to avoid falling. Much evidence has shown also the link between altered perceived verticality and clinical conditions that affect the vestibular system from the peripheral to the central processing of balancing information [16], as well as other clinical conditions, such as vestibular migraine [17], Parkinson's disease [18], multiple sclerosis [19,20], and brain stroke [21].

The Aubert effect or A-effect [22] is a well-known bias in the perception of verticality toward the head position that takes place when tilted to the side. This phenomenon has been investigated in uni- and multisensory conditions by focusing on the subjective visual vertical SVV [23–26], the subjective haptic vertical SHV [27–29] the subjective auditory vertical [30], and the interaction of visual and haptic sensory information on perceived verticality [27,31]. By asking subjects to indicate the direction of gravity when upright

or tilted to the side by aligning either a visual line or a haptic rod with gravity, several studies found that verticality judgments were biased toward the head position when subjects' tilt was greater than ~60°, indicating that the signal relative to head tilt is generally underestimated. In other words, judgments are attracted to the most common position assumed by the head, i.e., the upright position, leading to biases in verticality estimation in unusual (non-upright) conditions. Interestingly, the A-effect has also been suggested to influence aesthetic preference of visual lines [32]. Opposite to the A-effect, the E-effect (with "E" indicating Entgegengesetzt, that is "opposite" in German) is observed when verticality estimates are biased away from body tilt [33]. Such an effect has been observed for tilts of a few degrees [34] and tilts in the range between 135° and 150° [35,36].

Such perceptual phenomena were interpreted by Mittelstaedt [23] as the consequence of a head-fixed idiotropic vector that the brain uses as a compensating strategy to correct for an unbalanced distributions of sensory cells in the otolith organs. According to this view, perceived verticality is the resultant of the combination of two vectors, an earth vertical gravity vector and the idiotropic that is centered to the head. Thus, once tilted, the resultant percept is shifted away from the veridical direction of gravity towards the tilted head. In this sense, the idiotropic vector serves as a reference for verticality to decrease the probability of making errors at small tilts but it comes with the side effect of causing greater inaccuracies at large tilts, that is the A-effect. Later on, Eggert interpreted the A-effect in Bayesian probabilistic terms [37] with a prior centered at the head that influences the readout of sensory information coming from the vestibular organs; mathematically, the role of such prior is similar to the role of the idiotropic vector theorized by Mittelstaedt. More recently, de Vrijer and colleagues [34,38] proposed a Bayesian model based on the assumption that sensory signals encoding head tilt and ocular torsion are overall accurate but imprecise, due to sensory noise. Such a model well accounts both for A- and E-effects. Instead of considering head tilt estimates inaccurate, this Bayesian model decreases the sensory noise associated with the vestibular readout of head tilt, by computing the percept with prior knowledge and experience on head and eye position. Such construct corresponds to the upright position, considering that this is the position that we mostly assume when aligned to gravity. In other words, the A-effect can be thought as follows: when tilted to the side, perception of head tilt is affected by the unusual head orientation and tends to be biased towards the most common position assumed in daily life, that is upright. Although not systematically observed as the A-effect, the same Bayesian model of de Vrijer and colleagues explains the E-effect as the consequence of an overestimation of the ocular counteroll at small head tilts. In particular, considering that head estimation is noisy, and thus imprecise, at small tilts the less noisy ocular roll movement signal may be encoded with a greater magnitude compared to the head tilt, thus an overestimated ocular counteroll. It follows that, the estimation of head tilt is less salient due to wide sensory noise and thus the head signal, if solely based on this information, may be coded as un-tilted; in such scenario, ocular counteroll encoded as greater than the actual head tilt drives the signal leading to an overestimation of head and body orientation relative to gravity. In the case of larger tilts, the head in space deviation overcomes the ocular signal, thus leading to the A-effect. Such perceptual phenomenona can be considered as the consequence of the probability according to which a certain condition is more likely to take place based on previous experience. Following such a probabilistic approach, prior information (e.g., prior experience) can then affect accuracy, especially in unusual conditions, such that perception is not merely the sensory readout of the physical information, but a complex phenomenon that brings probability and previous experience into play.

A simplified version of Bayesian probabilistic models to predict the A-effect in the perception of verticality [34,38] is represented in Figure 4.

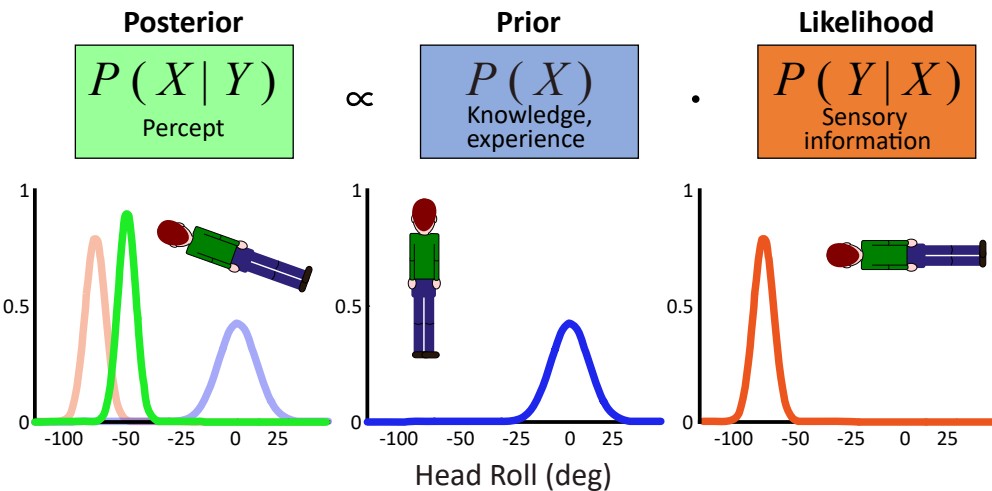

**Figure 4.** Representation of a simplified Bayesian model for the A-effect. The green curve describes the perceived head roll as a posterior probability distribution proportional to the product of the sensory information described by a likelihood function (orange curve) centered at the physical head roll (peak = $-90°$) and a prior distribution (blue curve) centered at the upright position (head roll = $0°$). The posterior (green curve) appears pulled toward the prior distribution (peak = $-50°$) and slightly narrower compared to the likelihood. All probabilities are assumed to be normally distributed.

In this framework, judgment of own body roll tilt is based on the conditional probability of observing a certain stimulus ($X$) given the sensory input ($Y$). In probabilistic terms, this scenario can be represented as follows:

$$P\left(X|Y\right) = \frac{P(Y|X)\,P(X)}{P(Y)}$$

Since *P(Y)* serves normalization purposes, the equation can be rewritten as follows:

$$P(X|Y) \propto P(Y|X)P(X)$$

where *P(X | Y) is* the posterior proportional to the product between the likelihood *P(Y | X)*, that is the conditional probability of observing the sensory input ($Y$) given the stimulus ($X$), and the prior distribution *P(X)*, which indicates a priori knowledge or experience about the occurrence of stimulus ($X$). In this example, the maximum a posteriori (MAP) rule is used to infer the perceptual estimate, according to which observer's judgment corresponds to the peak of the posterior distribution.

As the head is tilted to the side, sensory information transduced by the vestibular system can be represented as a likelihood distribution centered at head tilt (see Figure 4). The estimate depends on the posterior, the product between sensory information (likelihood) and the prior. If estimates were based exclusively on sensory information, no bias would be reported in head tilt perception, meaning that the prior distribution would be flat. On the contrary, in the case of the A-effect, a prior centered at the upright position (no tilt) well accounts for the bias in response to verticality judgments. The rationale behind this model is that the more common position of the head relative to gravity is upright. Thus, judgments of uncommon orientation are pulled toward the most probable one, causing underestimation of the head tilt.

The development of such a prior corresponding to the upright position may be related to the integration of multiple sensory cues via perceptual experience and redundancies. Although the readout of graviception is mostly vestibular-based as the vestibular system constantly reacts to changes of head orientation with respect to gravity, other sensory modalities definitely play a role in obtaining the association between verticality and the direction of gravity. Let's consider the orientation of windows, doors, or trees, unless

specific cases, these objects tend to be earth-vertical, that is aligned with gravity. Along these lines, gravity priors have been observed in the perception of moving object speed with higher precision for objects moving downward compared to upward [39]. However, our brain does not perceive external objects via the vestibular system but employs other sensory modalities, such as vision and haptics. In this context, it can be suggested that gaining an upright prior may be based on the association between vestibular input informing the brain about the direction of gravity and other sensory modalities processing objects' properties such as the orientation. Starting from this perspective, investigation on the A-effect across child development revealed how the influence of an upright prior on perceived verticality follows an age- and sensory modality-dependent path. Cuturi and Gori [31] tested children from 5 to 12 y.o. and adults in a discrimination task that allowed them to quantify perceived verticality when tilted to the side. In detail, the authors tested visual and haptic perception of verticality with children tilted 90° to their left-ear. The application of a Bayesian modeling approach revealed the presence of a consistent A-effect for all tested children and adults when verticality was tested visually. Interestingly, haptically perceived verticality showed an A-effect only in children and a progressive decay of the effect to be roughly absent in in adults. Bayesian modeling revealed that the upright prior changes across age, with children showing a more prominent presence of a prior centered at the upright orientation of the body both for visual and haptic verticality whilst in adults the upright prior is maintained only for the visual readout of verticality. The reason behind these sensory-dependent changes in the influence of the upright prior may rely on the properties of the development of the balance control system across age. The ontogenesis of the ability to move through space and maintain balance has been related to the disambiguation between head and body reference frames [40,41]. Kinematic studies across development show that when 6 y.o. children walk in challenging balancing condition, they increase their head-trunk stiffness (i.e., the "en-bloc" mode) whilst older children decorrelate head and trunk movements [40,42]. In other words, across development children go from an "en-bloc" mode of head and body coordination, i.e., processed as a whole unit, to an articulated coordination mode where head and body are disambiguated as two different instances. It can be hypothesized that haptic processing of verticality may receive the same influence from an upright prior as visual verticality up to the developmental momentum in which coordination of head and body is disambiguated in the articulated coordination of two different units. Supporting this view, haptic judgments of verticality have been linked to body rather than head reference [27]. As age increases, the disambiguation of head and body leads the brain to rely less on body-related sensory modalities, e.g., haptic for the perception of gravity, and instead confines this sensory system to the processing of object's orientation regardless of the orientation of the body relative to gravity. In other words, the developing brain may use both visual and haptic information to process and bind verticality with the direction of gravity. Then as head and body are processed as separate entities in the context of balance behavior (i.e., from around 8 y.o. [40,41]), the binding between vision and graviception remains. Vision is indeed the prominent sensory modality used to encode properties of the external word that involve the orientation of our body relative to the surroundings.

The contribution of other sensory modalities in perceived verticality has been studied also with somatosensory loss patients [43]. In particular, the presence of the A-effect has been shown to depend on the availability of residual somatosensory information, indicating that an online processing of proprioceptive information is fundamental to access the upright prior. However, it cannot be excluded that such a prior is absent, as hemiplegic patients still showed the A-effect when tilted to the non-paretic side. To better understand the ontogenesis of such a prior, combined studies on blind individuals and across development provide a model to test the specific role of visual experience in cross-sensory calibration for balancing control. The investigation of haptic perception of verticality in early and late blind adults probed the fundamental role of vision in multisensory perception by showing how visual experience shapes the presence of the above mentioned upright

prior [28]. Early blind individuals have no consistent biases in perceiving verticality when tilted clockwise except for those who perform echolocation to orient themselves during spatial navigation, whose responses, as in late blind subjects, show an A-effect. Interestingly, such an effect was not present in healthy sighted people who instead show the A-effect mostly in the visual modality. Together with the study on children, these findings indicate visual sensory information to be pivotal not only in gaining functional perception of object orientation [44,45], but also in influencing multisensory readout of vestibular information about head roll tilt indirectly measured through verticality judgments in unusual position. The development of an upright prior signaling the most important posture needed for spatial navigation might be based on the visual input during the early years of development. In the case of early blind individuals, it might be acquired with independent spatial navigation experience likely enhanced by echolocation behavior [46]. Body rather than head centered reference could lead late blind individuals to anchor their visually acquired upright prior to the haptic modality with no disambiguation between head and body references observed instead in sighted individuals across development. Supporting this view, the influence of priors does not seem to develop in those individuals who did not experience vision during early stages of development.

A recent study suggests that haptic judgments of verticality are sensitive to distortions that depend on hand related bias rather than graviception itself [47]. In the context of studies on blindness [28], because of the absence of vision, haptic information becomes prominent in the processing of object orientation. It can therefore be hypothesized that such hand-bias related distortions may simply be compensated by the brain in such population. At the same time, many body-related changes take place during growth, e.g., the hand itself changes in size, but such changes do not influence children's perception [48], thus suggesting that also in the context of developmental studies, compensation mechanisms may take place. Although methodologically challenging, future research may investigate haptic perceptual verticality by correcting for potential hand-related distortions to deepen the understanding of biases in graviception in the context of multisensory integration and sensory loss. For instance, recent research provided evidence for a biased perception of hand position in space when pitch-tilted to gravity [49]. In the context of the A-effect, such an investigation would also need to be extended to the roll plane.

## 3. Inaccuracies in Heading Perception

Direct measurements of vestibular perception often employ motion simulators that elicit passive self-motion stimulation, e.g., [11,50]. These instruments, combined with psychophysical methods have allowed researchers to discover the contribution of different sensory modalities in coding self-motion [13,51,52] and the involvement of brain areas that subtend properties of self-motion perception, such as the direction in which we are heading [53,54]. In the context of biases, studies on heading perception show a systematic overestimation for the oblique directions of movement in the horizontal plane both with discrimination [55] and identification tasks [55,56]. Within a Bayesian framework, such biases can be interpreted as the byproduct of a non-uniform prior. In detail, this has been identified with priors centered in the lateral directions, in other words heading biases are a byproduct of the categorization of movements as either leftward or rightward. As reported in the introduction, biases of this kind can be considered as evidence of perceptual expansion around the category boundary. In the case of heading perception, the category boundary corresponds to straight ahead, that is heading 0°. In turn, such a scenario leads to increased discriminability at the cost of biased judgments for intermediate directions (i.e., the oblique directions) toward lateral directions. In accordance with this view, repulsion from a category boundary has been suggested to interpret biases in perceptual phenomena other than heading during self-motion. In the context of biological motion perception, Sweeny and colleagues [57] observed that the evaluation of a walker's direction (i.e., point light walker), heading towards the observer, is systematically biased towards lateral directions, similar to the heading biases in vestibular perception. Moreover, in this case, this

pattern of results can be interpreted as the consequence of a repulsive effect centered at the category boundary, which corresponds to a walker heading $0°$, that is directly towards the observer, to avoid head-on collision. Taken together, these findings suggest that perceptual processing of space during navigation might integrate the overestimation observed for both heading processes to improve avoidance of approaching obstacles while moving through space.

Interestingly, vestibular heading biases have been confirmed also from a neural perspective. In detail, such biases were predicted with a population vector decoding approach considering the known overrepresentation of lateral directions of movement in brain areas processing heading stimuli, that is MSTd, (i.e., the dorsal medial superior temporal cortex) [54]. The non- uniformity of the distribution of preferred directions across neurons in MSTd reveals the nature of the estimates to be away from the category boundary (heading $= 0°$), thus linking computational and neural levels of explanation of the biases [55,56]. According to the population vector decoding approach, the brain interprets the physically presented stimulus based on the pattern of neural responses across the population (the encoding level) as a whole. As more units in MSTd prefer the lateral directions of movement, their activity will be more influential in judging heading. Consequently, oblique heading directions will be overestimated when predicted by a population vector decoder and judged away from straight ahead [54].

Studies on heading perception have revealed the presence of different pattern of biases depending on factors, such as body and stimulus orientation. Regarding the first, vestibular heading perception in supine position is susceptible to changes compared to the upright position, whilst visual heading accuracy is not affected by body orientation relative to gravity [58]. These findings indicate that the use of visual self-motion stimuli in contexts where participants are forced to be supine (e.g., in brain imaging studies) is preferrable over vestibular stimulation which, other than being methodologically challenging, may be affected by body orientation (see also the previous chapter in this review article). Regarding stimulus orientation, heading perception of movements on the earth-vertical plane, either coronal or sagittal to body position [59], show that in this condition heading biases in the vestibular domain are more pronounced compared to visual heading bias, a pattern that is instead observed in the investigation of heading perception in the earth-horizontal plane [55,56].

## 4. Vestibular and Cross-Modal Aftereffects

Aftereffects can be considered as biases induced by brain adaptation to steady changes in the environment. Therefore, the presence of aftereffects reflects the activation of brain networks underlying the perceptual dimension of interest (e.g., upward vs. downward visual motion). In the context of unisensory perception, much research has focused on the investigation of visual motion aftereffects, or MAEs to elucidate the different stages of motion processing in the visual cortex (for a review, [60]). Perceptual mechanisms behind aftereffects have been explained by Barlow in his "law of repulsion" [7], which postulates that a "repulsive force" between coupled perceptual mechanisms comes into play, leading to "decorrelated" activity. Thus, the brain avoids coding redundancy in signaling the familiar stimulus, such that novel stimulation can be better detected. In this context, the presence of a motion aftereffect in a sensory domain (e.g., auditory) consequent to adaptation in a different sensory domain (e.g., visual) would indicate the presence of a multisensory neural network in the tested perceptual dimension.

Regarding vestibular perception, the investigation of within-modality aftereffects has received less attention than in vision. The investigation of aftereffects in self-motion perception has shown the presence of a time window of approximately 3 s within which a passive self-motion stimulus influences accuracy in the perception of a subsequent stimulus [61]. From a methodological perspective, this finding has non-negligible consequences as it suggests maintaining an inter-stimulus-interval of at least 3 s in case of subsequent vestibular stimulation to avoid influence across stimuli. Similarly, the influence of subse-

quent self-motion stimuli of different nature, such as rotation and translation, has revealed canal-otolith interactions that lead to aftereffects [62]. The presence of such aftereffects indicates that, as with the visual system, the vestibular system shows direction specific responses that can be investigated with MAE alike paradigms. On the other hand, such a phenomenon provides behavioral evidence for the presence of neurons responding to directional vestibular stimulation in the human brain. Along these lines, vestibular aftereffects have been seen in the vestibular perception both in response to roll and yaw rotation. Regarding roll rotation, first observations were made in the context of aviation with the otherwise known "Gillingham illusion", which indicates an erroneous perception of roll following an abrupt roll, that is an aftereffect in the opposite direction of the adapter. Such a perceptual phenomenon has been shown both in flight simulation [63,64] and in less complex laboratory contexts [65] following short adaptation phases of 1.5–2 s. In the context of graviception, adaptation to prolonged roll-tilt of the head causes distortions in the internal estimate of verticality when both head and body are tilted relative to gravity [5]. When the head is tilted on the body with a roll tilt of about 15–20°, perceptual shifts in perceived verticality are stronger and interestingly not correlated with ocular torsion [3]. Considering these findings [3,5], the influence of adaptation to steady changes in head orientation may be mostly related to changes of the head orientation relative to the body rather than to gravity, with biases that are not related to ocular-reflex mechanisms but to processes involved in the perception of verticality. Regarding yaw rotations instead, adaptation to stimuli that resemble the most common rotation frequencies experienced while walking (i.e., frequencies near 1 Hz) induces aftereffects in self-motion perception suggesting that coding redundancy for such movements may be reduced to increase discriminability of deviations from subsequent movements with similar frequency [66]. In the context of the velocity storage mechanism [67], i.e., those oculomotor pathways that serve to prolong self-motion rotation beyond peripheral vestibular organs deactivation due to constant velocity movements, yaw aftereffects suggest that vestibular perception and vestibular reflex follow different mechanisms [66]. Supporting this view, Coniglio and Crane [66] observed that vestibular adaptation with a fixation point, which should decrease the time constant of the velocity storage mechanism, results in perceptual aftereffects, but not in persistent perception of self-motion. The latter would have been observed if the velocity storage mechanism had been activated; its absence indicates that the velocity storage might not be strictly linked to vestibular perception [66].

In the context of multisensory perception, aftereffects between sensory modalities provide an indistinctive measurement of multisensory interaction at the brain level via purely behavioral stimulation. Specifically, such paradigms unveil cross-modal interactions by testing whether adaptation in one sensory modality influences perception in another tested sensory modality. Much research has investigated such perceptual phenomena, revealing the presence of visual-auditory [68,69], audio-visual [70], visual-tactile [71] and visual-vestibular cross-modal aftereffects [72–74]. Such phenomena provide evidence of the presence of a shared perceptual space that links two different sensory modalities. Adaptation in one sensory modality suppresses neural response to the adapted stimulus such that the presentation of a neutral or ambiguous stimulus leads to a shift in perception in the opposite direction along the perceptual continuum. The presence of such a perceptual shift in the tested sensory modalities is thus evidence of a shared network between the investigated sensory modalities, which otherwise will not produce perceptual changes in the response to the test stimulus. In the context of visual-vestibular cross-modal aftereffects, the literature provides early reports of feelings of rotation after being adapted to an optokinetic stimulation, corroborated by the presence of afternystagmus as well [75]. Similarly, MAEs have been observed to be reduced when inertial motion was presented together with visual adaptation [76,77]. After a long hiatus in research, recent investigations of visual-vestibular cross-modal aftereffects pointed towards a more punctual identification of such perceptual phenomena. A preliminary investigation took advantage of a motion simulator and visual optic flow but did not reveal a neat transfer of adaptation between

the two sensory modalities [72]. Further research showed that to elicit aftereffects in the vestibular domain, a more sustained visual stimulus is needed [73]. In both researches, the investigation focused on the influence of an adapting stimulus in the visual domain (i.e., optic flow) and a test stimulus consisting in a passive linear translation on the earth-horizontal plane. Thus, a perceptual shift along the fore-aft continuum would indicate the presence of aftereffects. In [73], the authors employed an optic flow stimulus that overcomes vection latency (i.e., 7 s [78,79]) and thus provides a steady vection-eliciting visual stimulus to induce an aftereffect in the vestibular domain in the direction opposite to the adapted one. A vection stimulus is indeed perceptually preponderant as it can affect sound localization [80]. Similarly, a phenomenon of this kind has been observed with postural sway in response to adaptation to an optic flow stimulus [74], thus indicating an extension of cross-modal bindings when also involving proprioceptive information about body posture.

## 5. Conclusions and Future Directions

The works presented in this review outline how perceptual biases in the vestibular domain can unveil properties of the brain mechanisms that underlie the processing of graviception and self-motion perception. The verticality biases described in Section 2 provide an interesting path of research that unveils developmental properties of perception by pursuing an investigation of biases both across development and in blindness. This approach provides insightful findings on the fundamental role of the dominant sensory modalities across development in the context of the cross-sensory calibration theory [45]. In particular, the presence of vision during development is pivotal for the acquisition of perceptual abilities, such as auditory spatial representation [81] whilst the haptic sensory modality is fundamental for gaining size perception [82]. Nonetheless, research in vestibular perception still lacks such insights. For instance, it is known that visual loss impacts the ability to perceive vestibular yaw rotations [83] and that blind individuals may benefit from the integration of auditory [12] and haptic [84] cues to detect vestibular stimulation. At the same time, much research has shown developmental patterns in the emergence of spatial navigation abilities [85,86] and their integration with multiple sensory cues [87,88]. In the context of self-motion perception, ageing is known to influence the ability to integrate visual and vestibular cues [89]. However, the integration of studies on vestibular self-motion perception and developmental patterns represents a gap in the literature. Thus, an approach that combines the investigation of multisensory integration across development and the role of one dominant sensory modality has received less attention in the context of vestibular research. Along these lines, intrinsic biases (e.g., heading biases) and biases resulting from the adaptation to contextual stimulation (e.g., aftereffects) have mostly focused on adult and healthy participants with few exceptions, e.g., [90]. The investigation of such perceptual phenomena in the context of development and sensory loss may instead increase the understanding of how the brain processes and integrates vestibular inputs in relationship to changes in the environmental properties.

**Funding:** This research received no external funding.

**Institutional Review Board Statement:** Not applicable.

**Informed Consent Statement:** Not applicable.

**Data Availability Statement:** Not applicable.

**Acknowledgments:** Part of the material presented in this review article is adapted from the author's PhD thesis entitled "Intrinsic and Induced Biases in Self-Motion Perception". I wish to thank Paul MacNeilage and Monica Gori for the fruitful discussions.

**Conflicts of Interest:** The author declares no conflict of interest.

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
