# Peer review of "Perceptual Biases as the Side Effect of a Multisensory Adaptive System: Insights from Verticality and Self-Motion Perception"

_2411-5150, 2022_

Round 1

Reviewer 1 Report

Overall, this is a well written brief review but there are some inaccuracies in the perceptual processes described and the review is missing out on some relevant key findings from the literature. I think it would be helpful to define here what is exactly meant by the vestibular system? Our perceptual processes aim to extract information from a stimulus through the signals that can be from various sensors. So, if there is inaccuracy in these processes, is that because of inherent bias in a sensor or because of how multisensory integration is weighted towards a specific modality in certain situations? So which one is the culprit here? In some places ( e.g., page 3 line 110-112 or the title of the manuscript) , the author seems to put the blame on the sensor (vestibular system), but I don’t think based on the literature and what we know from the psychophysical studies, it can be distinguished where this inaccuracy exactly lies; i.e., multisensory perceptual processes or a specific system like the vestibular system? It would be very important to clarity this point to avoid forming notions that might not truly represent the source of perceptual inaccuracy in the big picture when it comes to the brain function. It would be more accurate to put everything in the context of multisensory integration, and that how one sensory modality in certain conditions may contribute more than other senses, depending on the task at hand, sensory noise/loss, etc. 

Figure 1, discriminability means discriminating between stimulus and reference or between two stimulus S1 and S2? 

Page 3 line 90: There are studies that have looked at adaptation of gravity perception with head tilt that could be properly cited here: 

https://www.frontiersin.org/articles/10.3389/fnhum.2016.00573/full

https://pubmed.ncbi.nlm.nih.gov/25185812/

https://journals.sagepub.com/doi/10.1080/14640747008401916

Page 3, Typo line 114: “ a car o flying”

Page 4, line 12: This is confusing. The semicircular canals are also part of the vestibular system, so the brain is still using vestibular signals to distinguish the two forces.  This is against the point made earlier in the former sentence 

Page 4, line 143: Not sure why it is relevant to point out specifically multiple sclerosis? The clinic relevance is about the location of the lesions and not the type of the lesions. For example, a stroke or MS lesion in the same brain location can have similar effect on gravity perception. It would be also helpful to cite other relevant clinical conditions: 

https://pubmed.ncbi.nlm.nih.gov/18678565/

https://pubmed.ncbi.nlm.nih.gov/20206672/

https://www.ncbi.nlm.nih.gov/pmc/articles/PMC6218433/

Page 4, lines 143-149: This is incorrect.  Idiotropic vector was first proposed by Mittelstaedt but it cannot account for the E effect, and thus not a correct reference to fully account for the processes involved in vertical perception. The Bayesian model that was later proposed can account for both A and E effects and systematic errors in vertical perception.  Also, this is just a model and it would be inaccurate to take it for the fact that “ the perception of verticality is accomplished via an idiographic vector”. The Bayesian model is actually described later in the text but unclear why the A effect was described in the context of idiotropic vector.  Here are some relevant citations: 

https://jov.arvojournals.org/article.aspx?articleid=2193491

https://www.ncbi.nlm.nih.gov/pmc/articles/PMC5660972/

At the end of this section, your reader may want to know how you would describe E effect in the Bayesian context of signal and prior integration? 

Figure 2 legend, and line 130: It is more appropriate to use pitch instead of tilt. 

Page 5, line 171: A more relevant reference here would be this paper by De vrijer et al: https://jov.arvojournals.org/article.aspx?articleid=2193491

Page 6, line 202: Again, idiotropic vector is different from multisensory Bayesian framework  and they should not be conceptually mixed up. 

Page 6, lines 215-221: The point raised here is not valid as haptic vertical and visual vertical tasks have inherent biases associated with their measurements (e.g., hand in space for the haptic task and eye in head for the visual task), and without considering these task biases and correcting for them, their results ( i.e., A or E effects) cannot be compared in a meaningful way.  See these relevant references: 

https://www.sciencedirect.com/science/article/abs/pii/S0306452221006047

https://www.jstor.org/stable/1421016?origin=crossref

https://journals.plos.org/plosone/article?id=10.1371/journal.pone.0145528

Page 6, lines 222-223: This sentence is unclear. What do you mean by “presence of prior-peaked”. Also, again, the concept of idiotropic vector is mixed up here with the Bayesian framework 

Page 6, lines 228-229: It is unclear where this statement comes from and what is evidence behind it?

Page 5, lines 232-240: Again, not sure how this claim can be justified based on the comment above on lines 215-221 ? 

Page 7, lines 250-253: Why just visual inputs? It would be more accurate to state that it is a multisensory process. There are several studies that have also showed the role of proprioceptive inputs. See this relevant reference for example: https://pubmed.ncbi.nlm.nih.gov/21097492/

Page 7, lines 253-256: Again, unclear what the isotropic vector is based on the comments above. 

Page 7, lines 260-262: This is unclear. What priors are you referring to? 

Page 7, lines 271-271: Why “ byproduct”? And not product? What do you mean by “ prior peak”? It would be helpful to make these sentences clearer. If forced choice tasks are used in these studies, the overestimation can be related to the method of measurement. Why not? 

Page 8, line 316: There are also several studies on vestibular aftereffects and gravity perception. See some references in the comment above on line 90, page 3. 

Page 8 lines 341-344: This unclear. It is helpful to clarify that the velocity storage mechanism is mainly a vestibulo-ocular process but it is hypothesized that it might also pertain to vestibular perception to facilitate low frequency motion perception. But why and how this mechanism has to be bypassed to generate the motion aftereffect? 

Page 9 lines 346-249: Suggest that you introduce the term ‘cross-modal interaction’ here as well. 

Page 9 lines 384-386: It is not clear what insights (or lack thereof) you are referring to here? 

Author Response

Reviewer 1

Overall, this is a well written brief review but there are some inaccuracies in the perceptual processes described and the review is missing out on some relevant key findings from the literature. I think it would be helpful to define here what is exactly meant by the vestibular system? Our perceptual processes aim to extract information from a stimulus through the signals that can be from various sensors. So, if there is inaccuracy in these processes, is that because of inherent bias in a sensor or because of how multisensory integration is weighted towards a specific modality in certain situations? So which one is the culprit here? In some places ( e.g., page 3 line 110-112 or the title of the manuscript) , the author seems to put the blame on the sensor (vestibular system), but I don’t think based on the literature and what we know from the psychophysical studies, it can be distinguished where this inaccuracy exactly lies; i.e., multisensory perceptual processes or a specific system like the vestibular system? It would be very important to clarity this point to avoid forming notions that might not truly represent the source of perceptual inaccuracy in the big picture when it comes to the brain function. It would be more accurate to put everything in the context of multisensory integration, and that how one sensory modality in certain conditions may contribute more than other senses, depending on the task at hand, sensory noise/loss, etc. 

I thank the reviewer for the thoughtful comments. By following their suggestions, I revised the text and made several changes and insertions. In particular, I inserted a description of the Vestibular system, please see highlighted inserted text  (lines 109-155) and novel figure 2. Moreover, considering this comment and following ones, I changed the title of the manuscript to “Perceptual biases as the side effect of a multisensory adaptive system: Insights from verticality and self-motion perception”. Please, find a detailed answer to the comments in this document. Major edits in the revised manuscript are highlighted in yellow.

R1.1

Figure 1, discriminability means discriminating between stimulus and reference or between two stimulus S1 and S2? 

Thanks for this comment, discriminability is considered in terms of ability to discriminate between two stimuli S1 and S2, I made this point clearer in the main text, please see lines 80-83

R1.2

Page 3 line 90: There are studies that have looked at adaptation of gravity perception with head tilt that could be properly cited here: 

https://www.frontiersin.org/articles/10.3389/fnhum.2016.00573/full

https://pubmed.ncbi.nlm.nih.gov/25185812/

https://journals.sagepub.com/doi/10.1080/14640747008401916

Thanks for this suggestion. I added the references in the manuscripts, see insertion at lines 93-95

R1.3

Page 3, Typo line 114: “ a car o flying”

Thanks for spotting this typo, it is now corrected.

R1.4

Page 4, line 12: This is confusing. The semicircular canals are also part of the vestibular system, so the brain is still using vestibular signals to distinguish the two forces.  This is against the point made earlier in the former sentence 

Thanks for spotting this lack of clarity. As highlighted in the text, this happens in the case of mainly linear movement, thus mostly the otolith organs are responsible for the encoding of self-motion stimulation. To make this point clearer, I know edited the text indicating that “Due to Einstein’s principle of equivalence, the otolith organs within the vestibular system are unable to distinguish these two forces”

R1.5

Page 4, line 143: Not sure why it is relevant to point out specifically multiple sclerosis? The clinic relevance is about the location of the lesions and not the type of the lesions. For example, a stroke or MS lesion in the same brain location can have similar effect on gravity perception. It would be also helpful to cite other relevant clinical conditions: 

https://pubmed.ncbi.nlm.nih.gov/18678565/

https://pubmed.ncbi.nlm.nih.gov/20206672/

https://www.ncbi.nlm.nih.gov/pmc/articles/PMC6218433/

Thanks for pointing this out. I now added relevant citations in the manuscript, please see insertion at lines 211-212

R1.6

Page 4, lines 143-149: This is incorrect.  Idiotropic vector was first proposed by Mittelstaedt but it cannot account for the E effect, and thus not a correct reference to fully account for the processes involved in vertical perception. The Bayesian model that was later proposed can account for both A and E effects and systematic errors in vertical perception.  Also, this is just a model and it would be inaccurate to take it for the fact that “ the perception of verticality is accomplished via an idiographic vector”. The Bayesian model is actually described later in the text but unclear why the A effect was described in the context of idiotropic vector.  Here are some relevant citations: 

https://jov.arvojournals.org/article.aspx?articleid=2193491

https://www.ncbi.nlm.nih.gov/pmc/articles/PMC5660972/

At the end of this section, your reader may want to know how you would describe E effect in the Bayesian context of signal and prior integration? 

Thanks for this suggestion. Please find the edited text at lines 230-262

R1.7

Figure 2 legend, and line 130: It is more appropriate to use pitch instead of tilt. 

Thanks for this suggestion, I corrected the text accordingly.

R1.8

Page 5, line 171: A more relevant reference here would be this paper by De vrijer et

al: https://jov.arvojournals.org/article.aspx?articleid=2193491

Thanks for this suggestion, I added the suggested reference.

R1.9

Page 6, line 202: Again, idiotropic vector is different from multisensory Bayesian framework  and they should not be conceptually mixed up. 

Thanks for this suggestion, I now made the point relative to the idiotropic vector clearer in the text.

R1.10

Page 6, lines 215-221: The point raised here is not valid as haptic vertical and visual vertical tasks have inherent biases associated with their measurements (e.g., hand in space for the haptic task and eye in head for the visual task), and without considering these task biases and correcting for them, their results ( i.e., A or E effects) cannot be compared in a meaningful way.  See these relevant references: 

https://www.sciencedirect.com/science/article/abs/pii/S0306452221006047

https://www.jstor.org/stable/1421016?origin=crossref

https://journals.plos.org/plosone/article?id=10.1371/journal.pone.0145528

I thank the reviewer for pointing this out. Although I believe the finding from Kim et al. is interesting and worth to be mentioned (see insertion at lines 375-380), much research (including Fraser et al 2015) drew fundamental findings on perceptual verticality by using a SHV task that is not dissimilar from the one used in Cuturi et al (2017, 2019). Moreover, the findings mentioned in the lines highlighted by the reviewer, are based on the application of the same paradigm across different populations (i.e. sighted, late and early blind adults as well as young children of different ages), where the main finding is relative to the differences among participants when tested with the same paradigm. Overall, I think that stating that results in haptic verticality “cannot be compared in a meaningful way” is rather abrupt and limited in the context of the present review, where the focus is not on the methodological artifacts in the investigation of haptic perception. Instead, I believe it to be appropriate pursuing future studies by taking such limitations in consideration. Indeed, I appreciate the value of this observation thus I included a paragraph discussing it (see lines 375-388). 

R1.11

Page 6, lines 222-223: This sentence is unclear. What do you mean by “presence of prior-peaked”. Also, again, the concept of idiotropic vector is mixed up here with the Bayesian framework 

Thanks for pointing this out. Please see edited text at lines 320-323

R1.12

Page 6, lines 228-229: It is unclear where this statement comes from and what is evidence behind it?

Thanks for this observation, I added relevant citations and explanation in the text, please see edits and insertions at lines 327-336

R1.13

Page 5, lines 232-240: Again, not sure how this claim can be justified based on the comment above on lines 215-221 ? 

Please, see my reply to the mentioned comment.

R1.14

Page 7, lines 250-253: Why just visual inputs? It would be more accurate to state that it is a multisensory process. There are several studies that have also showed the role of proprioceptive inputs. See this relevant reference for example: https://pubmed.ncbi.nlm.nih.gov/21097492/

Thanks for this suggestion, I know inserted the relevant citation in the indicated section (lines 347-355).

R1.15

Page 7, lines 253-256: Again, unclear what the isotropic vector is based on the comments above. 

Following reviewer’s comments, the idiotropic vector is now mentioned only when describing Mittelstaedt’s model.

R1.16

Page 7, lines 260-262: This is unclear. What priors are you referring to? 

The upright prior. I have edited the whole section to improve the clarity, please see the edited text. 

R1.17

Page 7, lines 271-271: Why “ byproduct”? And not product? What do you mean by “ prior peak”? It would be helpful to make these sentences clearer. If forced choice tasks are used in these studies, the overestimation can be related to the method of measurement. Why not? 

Thanks for this comment. Biases are byproduct in the sense that the main system uses upright prior to maintain balance;  in laboratory context, tested with unusual conditions, the Aubert effect comes as a byproduct of an optimal system that works to maintain balance. This is the main message of the review. This concept is expressed in the context of heading as follows:

“In the case of heading perception, the category boundary corresponds to straight ahead, that is heading 0°. In turn, such scenario leads to increased discriminability at the cost of biased judgments for intermediate directions (i.e., the oblique directions) toward lateral directions.”

Regarding prior peak, I meant centered at specific orientations, considering the potential confusion I changed the word with “centered”.

Heading biases have been reported both with forced choice and identification tasks, I added this information within the text.

R1.18

Page 8, line 316: There are also several studies on vestibular aftereffects and gravity perception. See some references in the comment above on line 90, page 3. 

Thanks for this suggestion. Please, see inserted text at lines 468-490

R1.19

Page 8 lines 341-344: This unclear. It is helpful to clarify that the velocity storage mechanism is mainly a vestibulo-ocular process but it is hypothesized that it might also pertain to vestibular perception to facilitate low frequency motion perception. But why and how this mechanism has to be bypassed to generate the motion aftereffect? 

Thanks for asking this clarification, I now inserted more details relative to the work of Coniglio and Crane (2014), based on their results on yaw aftereffect, they suggest that the velocity storage mechanism may not be involved in vestibular perception. This point is now made clearer in the text, see lines 490-495

R1.20

Page 9 lines 346-249: Suggest that you introduce the term ‘cross-modal interaction’ here as well. 

Thanks for the suggestion, I introduced the term as suggested.

R1.21

Page 9 lines 384-386: It is not clear what insights (or lack thereof) you are referring to here? 

Thanks for the observation. I now made this clearer in the text, please see insertion at lines 535-541

Reviewer 2

Thank you for the opportunity to review "Biases in vestibular perception as the side effect of an optimal adapting system" by LF Cuturi.  This manuscript provides a review of the perceptual biases within sensory modalities, specifically related to the vestibular system.  The author provides a focused review of the sensory integration focused on gravitoinertial perception inaccuracies, directional heading, and aftereffects.  Description of these three specific areas provides an overview of the integration of vestibular sensory information into overall orientation and motion perception.

Overall, I find this manuscript to be a wonderful overview of the current literature in this area.  As a whole, understanding of vestibular perception and sensory integration is rudimentary and significantly more work is needed - as the author notes - to highlight developmental effects, the complex multi-sensory integration expected, and the challenges with aging.  I found this manuscript to highlight the current understanding of vestibular sensory integration as well as note the areas of considerable limitation.

I thank the reviewer for the thoughtful comment, I am pleased to read that they appreciated the proposed manuscript. Given both reviewers’ comments, I have also changed the title of the manuscript to “Perceptual biases as the side effect of a multisensory adaptive system: Insights from verticality and self-motion perception”. Major edits in the revised manuscript are highlighted in yellow.

R2.1

The manuscript provides a thorough review with appropriate references.  This will provide the reader with an introduction to the importance of the relationship between the visual and vestibular systems in the above-noted topics.  I have not found any recent similar review.

Thanks.

R2.2

The figures provide nice visuals for the concepts associated with bias and the Bayesian model.  The concept of A-effect is challenging to visualize and I applaud the author for this clear contribution.

Thanks.

R2.3

With regards to bias, the E-effect is also a concern in terms of head tilt and vertical perception.  Given that most head tilts will be <60-70-degrees, the E-effect is likely to occur more often during everyday life.  Perhaps this is not as dramatic, but it is more commonly expected.  Would the author comment on why this was not included in the discussion in chapter 2?

I thank the reviewer for pointing this out, by following also reviewer 1 suggestion, I expanded the explanation of the E-effect.

R2.4

Regarding the introduction, this could perhaps use a bit of wordsmithing to clarify the topic area.  I found that it jumped right in to the main crux of the discussion without any preamble.  I assume that many readers of Vision may be unfamiliar with the vestibular system and concepts surrounding navigation.  While it is true that the vestibular system encodes self-motion, this concept may not be clear to the reader.  I suggest a small paragraph on the vestibular end organs/reflex pathways to help the reader understand the subsequent illusions.

I thank the reviewer for suggesting this, I also find value in the insertion of such information, therefore I expanded and reviewed the introduction to provide the reader with a more compelling knowledge of the  vestibular system and related perceptual processes, please see highlighted inserted text  (lines 109-155) and novel figure 2.

Reviewer 2 Report

Thank you for the opportunity to review "Biases in vestibular perception as the side effect of an optimal adapting system" by LF Cuturi.  This manuscript provides a review of the perceptual biases within sensory modalities, specifically related to the vestibular system.  The author provides a focused review of the sensory integration focused on gravitoinertial perception inaccuracies, directional heading, and aftereffects.  Description of these three specific areas provides an overview of the integration of vestibular sensory information into overall orientation and motion perception.

Overall, I find this manuscript to be a wonderful overview of the current literature in this area.  As a whole, understanding of vestibular perception and sensory integration is rudimentary and significantly more work is needed - as the author notes - to highlight developmental effects, the complex multi-sensory integration expected, and the challenges with aging.  I found this manuscript to highlight the current understanding of vestibular sensory integration as well as note the areas of considerable limitation.

The manuscript provides a thorough review with appropriate references.  This will provide the reader with an introduction to the importance of the relationship between the visual and vestibular systems in the above-noted topics.  I have not found any recent similar review.

The figures provide nice visuals for the concepts associated with bias and the Bayesian model.  The concept of A-effect is challenging to visualize and I applaud the author for this clear contribution.

With regards to bias, the E-effect is also a concern in terms of head tilt and vertical perception.  Given that most head tilts will be <60-70-degrees, the E-effect is likely to occur more often during everyday life.  Perhaps this is not as dramatic, but it is more commonly expected.  Would the author comment on why this was not included in the discussion in chapter 2?

Regarding the introduction, this could perhaps use a bit of wordsmithing to clarify the topic area.  I found that it jumped right in to the main crux of the discussion without any preamble.  I assume that many readers of Vision may be unfamiliar with the vestibular system and concepts surrounding navigation.  While it is true that the vestibular system encodes self-motion, this concept may not be clear to the reader.  I suggest a small paragraph on the vestibular end organs/reflex pathways to help the reader understand the subsequent illusions.

Author Response

Reviewer 2

Thank you for the opportunity to review "Biases in vestibular perception as the side effect of an optimal adapting system" by LF Cuturi.  This manuscript provides a review of the perceptual biases within sensory modalities, specifically related to the vestibular system.  The author provides a focused review of the sensory integration focused on gravitoinertial perception inaccuracies, directional heading, and aftereffects.  Description of these three specific areas provides an overview of the integration of vestibular sensory information into overall orientation and motion perception.

Overall, I find this manuscript to be a wonderful overview of the current literature in this area.  As a whole, understanding of vestibular perception and sensory integration is rudimentary and significantly more work is needed - as the author notes - to highlight developmental effects, the complex multi-sensory integration expected, and the challenges with aging.  I found this manuscript to highlight the current understanding of vestibular sensory integration as well as note the areas of considerable limitation.

I thank the reviewer for the thoughtful comment, I am pleased to read that they appreciated the proposed manuscript. Given both reviewers’ comments, I have also changed the title of the manuscript to “Perceptual biases as the side effect of a multisensory adaptive system: Insights from verticality and self-motion perception”. Major edits in the revised manuscript are highlighted in yellow.

R2.1

The manuscript provides a thorough review with appropriate references.  This will provide the reader with an introduction to the importance of the relationship between the visual and vestibular systems in the above-noted topics.  I have not found any recent similar review.

Thanks.

R2.2

The figures provide nice visuals for the concepts associated with bias and the Bayesian model.  The concept of A-effect is challenging to visualize and I applaud the author for this clear contribution.

Thanks.

R2.3

With regards to bias, the E-effect is also a concern in terms of head tilt and vertical perception.  Given that most head tilts will be <60-70-degrees, the E-effect is likely to occur more often during everyday life.  Perhaps this is not as dramatic, but it is more commonly expected.  Would the author comment on why this was not included in the discussion in chapter 2?

I thank the reviewer for pointing this out, by following also reviewer 1 suggestion, I expanded the explanation of the E-effect.

R2.4

Regarding the introduction, this could perhaps use a bit of wordsmithing to clarify the topic area.  I found that it jumped right in to the main crux of the discussion without any preamble.  I assume that many readers of Vision may be unfamiliar with the vestibular system and concepts surrounding navigation.  While it is true that the vestibular system encodes self-motion, this concept may not be clear to the reader.  I suggest a small paragraph on the vestibular end organs/reflex pathways to help the reader understand the subsequent illusions.

I thank the reviewer for suggesting this, I also find value in the insertion of such information, therefore I expanded and reviewed the introduction to provide the reader with a more compelling knowledge of the  vestibular system and related perceptual processes, please see highlighted inserted text  (lines 109-155) and novel figure 2.

Round 2

Reviewer 1 Report

The author has sufficiently addressed all previous comments. Thanks